# Pregnancy in Slaughtered Lambs and Sheep—A Cross-Sectional Study in Three Abattoirs in Switzerland

**DOI:** 10.3390/ani12101328

**Published:** 2022-05-23

**Authors:** Chiara Pagamici, Roger Stephan

**Affiliations:** Institute for Food Safety and Hygiene, Vetsuisse Faculty, University of Zurich, CH-8057 Zurich, Switzerland; chiara.pagamici@uzh.ch

**Keywords:** sheep, lamb, pregnancy, slaughter, fetus, abattoir

## Abstract

**Simple Summary:**

The slaughtering of pregnant livestock and its relevance to animal welfare has become an ethically controversial topic. In this study, the prevalence of sheep and lambs being slaughtered while pregnant in Switzerland was assessed as well as the stage of pregnancy and the life signs of the fetuses. Data collection was carried out over one year in three Swiss abattoirs. Overall, 7.6% of the female animals were pregnant at slaughter, and 25.5% of them were in the third trimester of pregnancy, where 81.1% of the fetuses showed signs of life, such as a heartbeat or umbilical artery pulsation. To assess the relevance of animal welfare, it is discussed whether fetuses feel pain and stress. Even though science disagrees as to whether fetuses are capable of feeling conscious pain, it cannot certainly be ruled out, which is why the slaughter of pregnant sheep and lambs should be minimized as much as possible.

**Abstract:**

The slaughter of pregnant sheep and goats is not restricted in Switzerland. The aim of this study was to assess the prevalence of pregnant sheep and lambs being slaughtered in Switzerland and to determine the state of gestation and vital signs of the fetuses in order to assess the need to take measures and raise awareness of this issue. The data collection was carried out from March 2021 to February 2022, comprising 115 days in three abattoirs. A total of 18,702 sheep and lambs were included in this cross-sectional study, and 8770 were female (46.9%), 663 of which were pregnant at slaughter (7.6%). The pregnancy rate varied by age category: 404 lambs (6.1%) and 259 sheep (11.9%) were pregnant. The highest pregnancy rate was found in winter (25.7%). Among the 663 pregnancies, more than a quarter were multiple pregnancies (28.2%). A total of 169 animals were in the third trimester of pregnancy (25.5%), where living fetuses were mainly found (81.1%). As it cannot be definitively ruled out that fetuses feel conscious pain, the data from this study underline that, from an ethical point of view, there is a need for action and that measures must be taken to reduce the number of pregnant slaughtered animals.

## 1. Introduction

In recent years the slaughter of pregnant livestock and its relevance to animal welfare has become an ethically controversial topic that has gained international media attention [1,2,3]. Switzerland has one of the strictest animal welfare laws worldwide. Nevertheless, the slaughter of pregnant livestock is not regulated by law [4], e.g., in contrast to Germany, which has banned the slaughter of cattle in the third trimester of pregnancy as of September 2017 [5].

In 2017, a working group led by the Swiss meat sector association Proviande developed the first industry solution in Switzerland to avoid slaughtering pregnant heifers and cows, introducing an obligation to declare pregnancy for heifers from the age of 18 months and for cows from 5 months after the last calving. The pregnancy must be declared on the accompanying document for cloven-hoofed animals. It was then revised in 2019, and since January 2020 there is an obligation to declare pregnancy for heifers from the age of 15 months. Furthermore, a fine in the amount of CHF 100 was applied for the unjustified slaughter of pregnant heifers and cows. In January 2022, the penalty was raised. Accordingly, a livestock farmer is deducted CHF 200 per animal that is found pregnant at slaughter without justification [6].

However, the slaughter of pregnant sheep and goats is not restricted in Switzerland. This means that numerous viable embryos and fetuses are wasted as slaughter by-products without prior stunning. This must be viewed as a major ethical concern, especially taking into consideration that fetuses might be able to experience pain or stress during the slaughter of the mother.

No current data are available on the slaughter of pregnant lambs in Switzerland, and information regarding pregnancies in the slaughter of sheep is limited.

The aim of this study was to assess the prevalence of pregnant sheep and lambs being slaughtered in Switzerland, making it possible to evaluate the current situation in order to assess the need to take suitable measures and raise awareness of this issue.

The data collection was carried out over the period of a year, which made it possible to determine the seasonality in pregnancies. A further objective was to determine the state of gestation of the fetuses, classifying them by trimester. Vital signs, such as the heartbeat, pulsation of the umbilical cord, spontaneous breathing, or a response to the interdigital reflex were also assessed, as one of the major ethical concerns regarding the slaughter of pregnant livestock is the capacity of the fetuses to perceive pain and stress.

The outcome of this work should form a basis to develop further action in order to reduce the slaughter of pregnant sheep and lambs.

## 2. Materials and Methods

### 2.1. Data Collection

As part of a cross-sectional study, data on the frequency of pregnancy in slaughtered sheep were collected in three Swiss slaughterhouses. The slaughterhouses were chosen based on their high yearly numbers of slaughtered sheep. According to Proviande, a total of 240,711 sheep and lambs were slaughtered in 2020 in Switzerland [7].

The data collection period was from March 2021 to February 2022. Ten days of data collection per season and slaughterhouse were carried out in spring, summer, and autumn, and five data collection days per slaughterhouse were carried out in winter, making the data collection period a total of 115 days.

Sampling took place directly at the production line during the slaughterhouses’ working hours. For all slaughtered sheep, the age group (adult sheep, lamb, or milk lamb) was recorded. Adult sheep were labelled as such by the slaughterhouse if they had at least two permanent incisors. All sheep and lambs that were slaughtered during the data collection period were included in the data collection, except for the milk lambs. The age class was also listed on the taxation list of the slaughterhouse, which was printed at the end of every data collection day.

The gender was determined after seeking out the genital tract. The uteri of all female sheep were systematically examined optically and by palpation for pregnancies. In the case of a possible enlargement or asymmetry, the uteri were carefully dissected. If there was amniotic fluid, the embryos or fetuses were searched for and taken out. It was evaluated whether they showed any signs of life (heartbeat, pulsation of the umbilical cord, breathing, or a response to the interdigital reflex). The umbilical cord was then cut in order to allow the living embryos and fetuses to die as soon as possible. Furthermore, all found embryos and fetuses were photographed. Then, the list with the slaughter data, including the age group of the sheep, was printed and compared to the detected data.

No sheep were killed for the purpose of providing samples. There was no ethical approval required for this study.

### 2.2. Evaluation of the Embryos and Fetuses

For all embryos and fetuses, an evaluation was carried out, which included the measurement of the crown–rump length, the weight, the gender of the fetuses if already visible, and the anatomical development, such as the presence of body hair. Multiple pregnancies were noted, and each embryo or fetus was evaluated individually. Based on the obtained data, each fetus was assigned to a pregnancy trimester. The classification was carried out based on Habermehl (1975), using the crown–rump-length and the anatomical development [8]. Briefly, with a crown–rump length between 5 cm and 27 cm and as soon as the gender can be identified, the fetuses were classified in the second trimester of pregnancy, whereas if the crown rump length is 27 cm or more and the *Sinus inguinalis* is developed, the fetuses were classified in the third trimester of pregnancy.

### 2.3. Statistics

A descriptive analysis was used to summarize the collected data. Significant differences between groups were identified by a chi-square test using GraphPad Prism (Version 9.3.0 (345), GraphPad Software, San Diego, CA, USA), considering *p* values < 0.05 to be statistically significant.

## 3. Results

### 3.1. Total Pregnancy Prevalence in Slaughtered Sheep and Lambs

A total of 18,702 sheep and lambs were included in this cross-sectional study lasting from March 2021 to February 2022. Data collection was performed in three slaughterhouses: 7368 sheep and lambs were examined in slaughterhouse A, 6425 in slaughterhouse B, and 4909 in slaughterhouse C. Data collection was performed on a total of 115 days. Ten days of data collection per slaughterhouse were carried out in spring, summer, and autumn. Data from 5978, 4613, and 5245 sheep and lambs were collected in each season. In winter, five data collection days were performed per slaughterhouse, resulting in a total of 2866 sheep and lambs.

In total, 8770 of all the documented sheep and lambs were female (46.9%), 663 of which were recorded as pregnant at time of slaughter (7.6%). The pregnancy rate of female sheep and lambs differed depending on the season. The lowest pregnancy rate was found in summer (1.8%), followed by spring (3.5%) and autumn (7.2%), whereas the highest prevalence was found in winter (25.7%) (Figure 1). Pregnancy rates were significantly different between all seasons (chi-square 774.9, *p* < 0.05).

### 3.2. Pregnancy Prevalence in Slaughtered Sheep and Lambs, According to Age Category

In total, 16,362 of the slaughtered animals were lambs (87.5%), and 2340 were adult sheep (12.5%). The majority of the lambs were male (9764, 59.7%), while in adults, the majority were female sheep (2172, 92.8%). The pregnancy rate varied by age category: 404 of the female lambs (6.1%) and 259 of the female adult sheep (11.9%) that were slaughtered throughout the year were pregnant. This difference was statistically significant (chi-square 77.9, *p* < 0.05) (Table 1).

### 3.3. Multiple Pregnancies, Life Signs and Gestational Stage of the Embryos and Fetuses

Among the 663 pregnant sheep and lambs, 476 had single pregnancies (71.8%), whereas more than a quarter had multiple pregnancies (28.2%). Accordingly, 176 twins (26.5%) and 11 triplets (1.7%) were detected. Therefore, a total of 861 embryos and fetuses were found and documented. The multiple pregnancies varied depending on the age class of the mother. Whereas 83.0% of the lambs had a single pregnancy only 54.1% of the adult mother sheep carried just one embryo or fetus. More multiple pregnancies were found in adult sheep: 42.8% of them had twins and 3.1% triplets, compared to 16.3% with twins and 0.7% with triplets in lambs.

In 351 (52.9%) of all pregnancies, the embryos or fetuses showed signs of life, such as a heartbeat or the pulsation of the umbilical cord. Thus, they were alive at the time of evisceration. However, no positive reflex response to pressure exerted on the interdigital space was observed in any of the fetuses. Moreover, none of them showed spontaneous breathing after opening the uterus.

Based on the crown–rump length and the anatomical development of the embryos and fetuses, their ages were estimated. The stage of pregnancy was divided into trimesters. Of the 663 documented pregnancies, 128 were in the first trimester of gestation (19.3%), 366 were in the second trimester of gestation (55.2%), and 169 were in the third trimester of gestation (25.5%). The gestational stage of adult sheep and lambs was compared. Whereas, in both, more than half of the pregnancies were found in the second trimester of pregnancy (54.9% and 55.4%), more pregnancies in the third trimester were found in adult sheep (30.4%) than in lambs (22.4%).

Fetuses showing signs of life were found mainly in the third trimester of pregnancy, where the heartbeat or the pulsation of the umbilical cord could be observed in 81.1% of the pregnancies, followed by the second trimester of pregnancy (56.3%). The lowest prevalence of pregnancies with fetuses showing life signs was found in the first trimester of gestation (6.3%).

### 3.4. Prevalence of Pregnancies Based on the Slaughterhouses

The prevalence of pregnant female animals varied depending on the slaughterhouse. The lowest pregnancy rate was observed in slaughterhouse B, where 5.7% of pregnant sheep and lambs were found, followed by slaughterhouse A (8.1%). The highest prevalence was found in slaughterhouse C with 9.3% of the female animals being pregnant when slaughtered.

### 3.5. Prevalence of Pregnancies Based on Livestock Owners

Based on the data collection period in autumn, it was examined how pregnancies are distributed among the livestock owners of the slaughtered sheep and lambs. During the thirty data collection days in autumn, data from a total of 5240 slaughtered sheep and lambs from 445 livestock owners were recorded. A total of 2670 sheep and lambs were identified as female (51.0%), and 193 of them were pregnant when slaughtered (7.2%). In total, 356 owners (80.0%) brought only non-pregnant animals to slaughter, whereas the remaining 89 (20.0%) were accountable for the documented pregnancies. About one third of the livestock owners with pregnant animals (34.8%) showed a pregnancy prevalence of less than 10.0%, while a little less than two third of them (65.2%) had at least 10.0% of their animals pregnant when brought to slaughter.

## 4. Discussion

### 4.1. Prevalence Assessment of Pregnant Slaughtered Sheep and Lambs

This study showed that pregnancies in slaughtered sheep and lambs are occurring regularly in Switzerland, with an overall prevalence of 7.6%. It was observed that the pregnancy rate varied by age class. This might be due to the fact that not all lambs reached sexual maturity at the time of slaughter. However, it is notable that in slaughtered lambs pregnancies are also found on a regular basis. This is due to the definitions of adult sheep and lambs in the slaughterhouse, which classifies an adult sheep as such if it has at least two permanent incisors. The eruption of the incisors happens, depending on the breed, between the ages of 12 and 23 months [7], whereas the age of puberty is influenced by a number of internal and external factors, such as the breed, nutrition, and the season of birth. For ewe lambs, it also differs depending by the presence of the ram [9]. The attainment of puberty is reached by ewe lambs between the 5th and 10th month of life, whereas in ram lambs, it occurs at 3 to 6 months [10]. Therefore, between the ages of 5 to 23 months, the ewes identified as lambs at the slaughterhouse might have already reached sexual maturity and can consequently carry one or multiple embryos or fetuses at the time of slaughter.

### 4.2. Pregnancy Rate Depending on the Season

Pregnancy rates were significantly different between all seasons (spring 3.5%, summer 1.8%, autumn 7.2%, and winter 25.7%), with rates being highest in winter. This was to be expected, as many sheep breeds are seasonally polyestrous with the mating season in autumn. However, some breeds are less seasonal and breed year-round [11,12], which is why pregnancies were found in all seasons.

### 4.3. Prevalence of Pregnancies Based on the Livestock Owners

During data collection, variations in the occurrence of pregnancies were observed throughout the day. Batches with many pregnant animals were noticed, followed by batches with lone or no pregnancies. Therefore, the distribution of the pregnant slaughtered animals among the livestock owners was further examined using the data collection period in autumn. The fact that 80.0% of the livestock owners did not bring any pregnant sheep or lambs to slaughter shows that avoiding the slaughter of pregnant animals is feasible. It indicates that the overall pregnancy prevalence is caused by only certain livestock owners and therefore lies at the producer-level and is farm-specific. It can be assumed that these farms either keep the ram in the herd all year round or do not separate the ram lambs from the female animals before they reach sexual maturity.

### 4.4. Prevalence of Pregnancies Based on the Slaughterhouses

Notably, slaughterhouse B (5.7%) had a lower prevalence of pregnancies than slaughterhouses A (8.1%) and C (9.3%). Slaughterhouse B was also the smallest one, and livestock owners mostly brought a lower number of sheep and lambs to slaughter than in the other two. Having a lower number of animals might result in more accurate awareness of their pregnancies.

This suggests, again, that the slaughter of pregnant sheep and lambs is a farm-specific concern, which is lying at the producer-level and could be further lowered.

### 4.5. Estimation of the Age of the Fetuses

There are many parameters that can be used to assess the ages of sheep fetuses. Michel (1995) names the crown–rump length (CRL) and weight as the main parameters for age determination [12]. This approach was used in the data collection of the present work, using the CRL and adding the external fetal development, such as the degree of hair growth, to determine the age of the embryos and fetuses. However, McGeady et al. (2017) pointed out that in this context it is not possible to determine the exact ages of the fetuses since the CRL is influenced by several factors, such as breed, diet, and litter size [13].

The age assessment of the fetuses was first carried out in weeks, based on the classification scheme of Habermehl (1975) using the recorded crown–rump length and the anatomical development, such as the presence of body hair. Then, the average of the age classification of the two parameters was calculated, as they did not always correspond to each other. The resulting age was assigned to the respective pregnancy trimester. As Habermehl (1975) did not state a CRL for every gestation week, in some cases it was necessary to assign a timeframe of more than one week to the corresponding CRL. This led to less accurate age estimation results when gestation was measured in weeks. By classification into pregnancy trimesters, this was avoided.

### 4.6. Distribution of Trimesters

Most fetuses were in the second trimester of pregnancy at time of slaughter (55.2%), and more than a quarter were found in the third (25.5%). The advanced age of the found fetuses emphasizes the ethical and animal welfare significance. During data collection, it could be observed that pregnant uteri were already visually identified in an early stage of gestation, including in the first month of pregnancy.

More pregnancies in the third trimester were found in adult sheep than in lambs. This might be explained by the fact that compared to the adult sheep not all lambs reached sexual maturity early enough to be in the third trimester of pregnancy at slaughter.

### 4.7. Multiple Pregnancies in Sheep and Lambs

The results from the data collection revealed that adult sheep were more likely to have twins or triplets than lambs. This coincides with the statements by Rüsse and Grunert (1993) that primiparous and multiparous sheep have a higher probability for multiple pregnancies than nulliparous animals [14]. Therefore, the risk of losing multiple fetuses is higher through the slaughter of an adult ewe than through the slaughter of a lamb. It has to be considered that multiple pregnancies in lambs were found on a regular basis, which also corresponds to a higher financial loss. In conclusion, from an economic point of view especially, adult ewes should be examined for pregnancy before being sent to slaughter. 

### 4.8. Awareness and Pain Perception in Fetuses

To assess the relevance of animal welfare regarding pregnant sheep and lambs sent to slaughter, it is required to know whether fetuses feel pain or stress during slaughter, as they are not stunned. During data collection at the slaughterhouses, the interdigital reflex was tested in fetuses showing an umbilical artery pulsation or heartbeat. None of the fetuses showed a defensive reaction. However, that does not rule out the presence of a pain sensation in living fetuses.

Pain is defined by the International Association for the Study of Pain (IASP) as ‘An unpleasant sensory and emotional experience associated with, or resembling that associated with, actual or potential tissue damage.’ [15], whereas stress was defined by Johnson et al. (1990) as a threatened homeostasis of the body, leading to adaptive behavioral and physical changes [16]. Both can lead to the activation of endocrinological systems, such as the hypothalamic–pituary–adrenal axis [17,18,19], which is why in the literature the pain- and stress-reaction are often used as synonyms. This shows that the two phenomena cannot clearly be separated from each other. Another decisive element in the process of experiencing stress and pain is the ability to be aware. Awareness or consciousness includes three main components, vigilance, mental contents, and selective attention [20], and was defined by Brain (1961) as ‘To be conscious is to be aware of things, and the things may be objects outside ourselves, or our own memories, thoughts and feelings’ [21].

As defined by Lowery et al. (2007), conscious pain perception requires a nociception as the sensation of the stimuli and a perception with an emotional reaction, which are processed by the brain. Peripheral pain receptors, connectivity of the sensory nerves to the spinal cord, a functional spinothalamic tract, and thalamocortical connections are the anatomical structures that are necessary for the sensation of pain and for the capacity of sentience [18]. They are developed just after mid-pregnancy [22].

To reach an emotional perception, an intact neurophysiological function and a response of the fetus to the slaughter condition are necessary [19]. Since Mellor et al. (2005) claim it is not possible to be asleep and conscious [22], it is necessary to define whether the fetus is awake. The neurophysiological function can be examined by electroencephalography (EEG) and through measuring the endogenous neuroinhibitors. By late gestation, 95% of the time there are two sleep-states in the EEG of the fetus, which are the rapid-eye-movement (REM) or active sleep and the non-rapid-eye-movement (NREM) or quiet sleep, characterized, respectively, by a low-voltage high-frequency and a high-voltage low-frequency EEG. This suggests that 95% of the time the fetus is asleep and, thus, unconscious [23]. In between the two sleep states, a so-called awake state occurs. However, this fetal wakefulness is controversial. Rigatto et al. (1986) observed an ovine fetus in utero for 5000 h through a Plexiglas window. Signs of wakefulness, such as eye opening or purposeful movements of the head, were never observed [24]. Moreover, it was shown that neonates with anencephaly can still have EEG activity [25] and unconscious patients may have wakeful EEGs [26], which indicates that wakefulness alone is not enough to establish awareness.

Endogenous neuroinhibitors, such as adenosine and prostaglandin D2, are sleep-inducing, whereas allopregnanolone and pregnanolone are sedative. They therefore promote sleep and suppress fetal consciousness. Adenosine increases in hypoxemia, including during the slaughter process when the ewe bleeds out [27]. Hence, the mentioned suppressors contribute to keeping the fetus asleep and unconscious.

To summarize, according to Mellor et al. (2005), it is unlikely that fetuses can consciously feel pain and stress. Bellieni and Buonocore (2012), on the other hand, assume that fetuses have the ability to experience pain and distress from the second half of pregnancy. They observed that human fetuses reacted to certain stimuli on the skin from as early as ten weeks of pregnancy. Furthermore, they suggest that fetuses can be awakened by external stimuli and that the neuroinhibitors not being anesthetics but sedatives does not prevent the fetuses from feeling pain [28]. Moreover, hormonal and behavioral signs of pain in fetuses can be recognized. A change in the heart rate [29], a rise in stress hormones, such as cortisol, endorphin, and norepinephrine [30], and a hemodynamic stress response with an increase in pulmonary vascular resistance [31] are seen as stress reactions of the fetus. However, it is unclear whether these stress reactions are consciously perceived.

In summary, science disagrees as to whether a conscious pain sensation occurs in fetuses. However, it cannot be certainly ruled out that fetuses feel pain or stress during slaughter. Therefore, from an animal welfare point of view, the slaughter of pregnant farm animals from mid-pregnancy should be reduced as much as possible.

Finally, the possible absence of pain in the fetus does not ethically justify the slaughter of sheep at an advanced stage of pregnancy without any reason.

## 5. Conclusions

This study represents the first systematic data collection regarding pregnant slaughtered sheep and lambs in Switzerland. The results prove that not only the slaughter of pregnant sheep but also the slaughter of pregnant lambs occurs regularly. The higher proportion of advanced pregnancies as well as the increased amount of living fetuses underline the ethical importance. That a fifth of the farmers were responsible for all the slaughter of pregnant animals implies that the issue is farm-based and can be solved at the producer-level. As long as the suffering of fetuses during slaughter cannot be certainly ruled out, the slaughter of pregnant sheep and lambs at an advanced stage of gestation has to be seen as an animal-welfare-related issue and should be avoided as much as possible, taking preventive measures, such as the optimization of herd management, the castration of male lambs, and pre-slaughter pregnancy controls in sexually mature animals.

## Figures and Tables

**Figure 1 animals-12-01328-f001:**
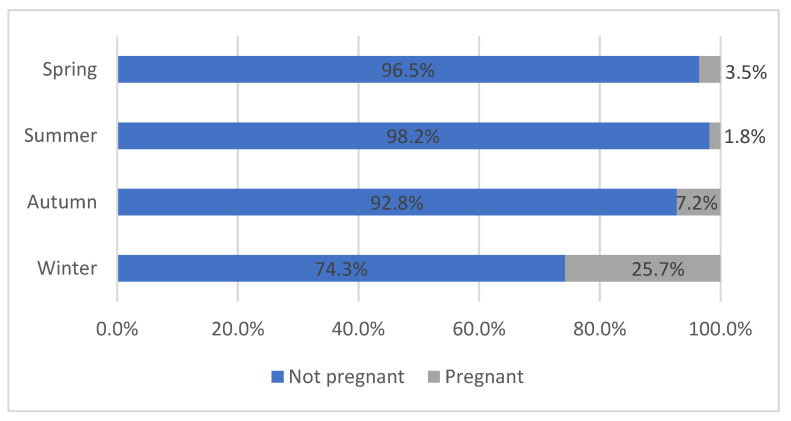
Pregnancy prevalence of female sheep and lambs at slaughter depending on the season.

**Table 1 animals-12-01328-t001:** Number of slaughtered sheep and lambs, divided by season, age class, gender, and pregnancy status.

	Lambs				Adults			
	Male	Female			Male	Female		
Season		Total	Pregnant	Not Pregnant		Total	Pregnant	Not Pregnant
Spring	3135	2098	50 (2.4%)	2048	61	684	48 (7.0%)	636
Summer	2577	1489	8 (0.5%)	1481	20	527	29 (5.5%)	498
Autumn	2526	1901	81 (4.3%)	1820	49	769	112 (14.6%)	657
Winter	1526	1110	265 (23.9%)	845	38	192	70 (30.4%)	122
Total	9764	6598	404 (6.1%)	6194	168	2172	259 (11.9%)	1913

## Data Availability

Data supporting the reported results are available from the authors upon reasonable request.

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
