# Peer review of "Pregnancy in Slaughtered Lambs and Sheep—A Cross-Sectional Study in Three Abattoirs in Switzerland"

_animals, 2022, doi:10.3390/ani12101328_

Round 1
Reviewer 1 Report
The manuscript makes a complete story, and presented good cases to the study of animal welfare. However, there are several comments and suggestions listed below.
- “Ten days of data collection per season and slaughterhouse were carried out in spring, summer and autumn, and five data collection days per slaughterhouse were carried out in winter.” Why not choose ten days for each season.
- Are there any regulations on slaughtering pregnant sheep and lambs in Switzerland? if possible, the author may add relevant regulations, that will enable readers to increase their understanding of local animal welfare.
- In the result section, the author may split Table 1 into multiple parts based on different topics, which will be clear for understanding.
Author Response
Reviewer 1
The manuscript makes a complete story, and presented good cases to the study of animal welfare. However, there are several comments and suggestions listed below.
- “Ten days of data collection per season and slaughterhouse were carried out in spring, summer and autumn, and five data collection days per slaughterhouse were carried out in winter.” Why not choose ten days for each season.
Answer: Based on the first three data collection periods in spring, summer and autumn, we have seen that the five additional sampling days (day 6 - day 10) are not changing the overall results per sampling period. Therefore, and due to time restrictions of the whole project, we have decided to integrate 5 sampling days per slaughterhouse in the winter period.
- Are there any regulations on slaughtering pregnant sheep and lambs in Switzerland? if possible, the author may add relevant regulations, that will enable readers to increase their understanding of local animal welfare.
Answer: There is a statement in line 51: “However, the slaughter of pregnant sheep and goats is not restricted in Switzerland.”
- In the result section, the author may split Table 1 into multiple parts based on different topics, which will be clear for understanding.
Answer: We simplified Table 1 taking away the division in slaughterhouses and making it clearer for understanding (see L132)
Reviewer 2 Report
General comments: This study aimed to determine the percentage of pregnant ewes slaughtered in 3 Switzerland slaughterhouses according the stage of pregnancy, age of females, season and the farmers clients. The ultimate goal is to discuss the ethical aspects of pregnant ewes slaughtering and characterize this situation given some preventive measures. To accomplish this aim, an observational study in slaughterhouse conditions, euthanizing the detected alive fetuses as soon as possible, was performed.
The subject is very relevant and is done as a descriptive study. This approach is a significant limitation of the study that contrast with the discussion section when some comparative terms were used (e.g., higher). This limitation can be easily avoided applying an inferential statistical analysis to the study. Also, several issues reported in the specific comments should be clarified or improved.
Specific comments:
L61-66: The main aim of the study is to identify the proportion of pregnant sheep at slaughter time. Secondly, to determine the seasonality of the pregnancies. The definition of pregnancy identification and its classification on 3 thirds should be placed in M&M.
L75: 11 months? See L108.
L75-78: Probably these days are selected according the slaughterhouses’ management and number of sheep. I suggest to well define and justify the selection of the sample. Can also the authors insert the official total number of annual reports for sheep slaughters in Switzerland?
L107-108: For M&M
L117-121: Please don’t repeat results. The figure 1 can be displayed as columns only with the percentage of pregnant ewes. Also apply a non-parametric test to evaluate where the differences between seasons are significant
L123: Why you report the males? The aim is to evaluate the proportion of pregnant females. I suggest to differentiate pregnant adult from pregnant lamb’s female and test if the differences were significant between them. Also, why you report each slaughterhouse individually? This is not relevant for this study (the main relevance is if your total sample is representative or not of the total population slaughter during one year in your country).
L136-143: If you wish to report the single vs multiple pregnancy according to age, please test the data according lamb vs adult and single vs multiple pregnancies (e.g., 4 outcomes groups in a table.
L144-148: This is mainly right for the 2nd and 3rd third of pregnancy. 150 day /3 = 50: The first right mainly included the embryo phase of pregnancy (not the fetal phase). Please report in M&M this difference in your classification and how to specific detect it (presence of embryo vesicle and embryo). The presence of umbilical cord pulsation and reflexes are dependent of the speed of evaluation. Also, I can understand that the authors insert this evaluation to confirm the ethical of the evaluation; but it was not observed in all the ewes. I think that an average interval between the slaughter and the evaluation should be reported as well as the absence of signs of a non-immediate death of the fetuses (e.g., mummifications/others).
L150. Not in trimesters (is not a cow) but estimated in third of gestation.
L149-155: Please use an inferential statistical analysis.
L162-167: What is the goal of this information? Do you wish to test differences between regions?
L169-173: This is a repetition of data.
L183: 9.6%?
L185: You need to test the values. If you wish to use the terms “twice as likely”, please perform an Odds Ratio analysis/presentation in results.
L191-198: Attention, for goats birthing at the breeding season, the manifestation of puberty (natural oestrus cycles) only occurs in next breeding season; for several of those occurring during middle and later anestrous season (about May-June to August), this manifestation probably occurs in the second breeding season (about 14-18 month later) when they are mated. Please turn clearer this part of the discussion, and add some additional references, once they can occur a great variability.
L195: The most part of the males reach puberty at 6-8 month, no? Please see the comment for females.
L200-203: Please see my comment regarding the OR.
L212-217: About 12 farmers presented more than 10%. One doubt: during non-breeding season, what is the percentage of sheep that can ovulate naturally? Or these ewes were subjected to estrous synchronization? Probably a discussion about this aspect can better explain some of your results.
L218-226: “Having a lower number of animals might result in more accurate awareness of their pregnancy”. The differences between slaughterhouses B e C are not evident.
L228-244: I don’t understand why you discuss this subject. You need to well define the estimation of the 1st, 2nd and 3rd pregnancy period in M&M. Some of this present information can be considered at definition time.
L246: “second pregnancy trimester” is not a trimester. Please correct in whole the text.
L245-253: I think that is important to discuss the pregnancy diagnosis in flock (slaughterhouses?) by echography, once is a quick and easy tool. In the 3rd pregnancy period, the pregnancy also can be diagnosed by abdominal succussion.
L257: “…that the older a sheep is, …”. Sorry but no entirely true: nulliparous ewes carry mainly single fetus; primiparous and multiparous females have more proportion of twins/multiple fetuses up to an old age/parity limit. The real (relevant) differences are between nulliparous and other parities.
L266-268: The interdigital reflex is a defense mechanism. I think that is relevant to report the interval between the slaughter of the mother and the evaluation of the fetus. Once the fetus is deprived of O2 and accumulate CO2, a progressive fetal depression occurs.
L300: A bovine fetus?
L325-326: Right. And at the current technology to make an easy and economic pregnancy diagnosis. There is no justification to slaughter pregnant females. It is nor ethical neither environmental (direct and indirect energy cost of a pregnancy) appropriate.
L331: The higher proportion.
L336-338: These preventive measures fit better if discussed in a subsection of the discussion section.
Author Response
Comments and Suggestions for Authors
General comments: This study aimed to determine the percentage of pregnant ewes slaughtered in 3 Switzerland slaughterhouses according the stage of pregnancy, age of females, season and the farmers clients. The ultimate goal is to discuss the ethical aspects of pregnant ewes slaughtering and characterize this situation given some preventive measures. To accomplish this aim, an observational study in slaughterhouse conditions, euthanizing the detected alive fetuses as soon as possible, was performed.
The subject is very relevant and is done as a descriptive study. This approach is a significant limitation of the study that contrast with the discussion section when some comparative terms were used (e.g., higher). This limitation can be easily avoided applying an inferential statistical analysis to the study.
Answer: Thanks for the comment. Statistical analysis was added to the study.
Specific comments:
L61-66: The main aim of the study is to identify the proportion of pregnant sheep at slaughter time. Secondly, to determine the seasonality of the pregnancies. The definition of pregnancy identification and its classification on 3 thirds should be placed in M&M.
Answer: The definition of pregnancy identification is already stated in M&M. L87 – 89: “In case of a possible enlargement or asymmetry, the uteri were carefully dissected. If there was amniotic fluid, the embryos or fetuses were searched for and taken out.” So is the classification on 3 thirds, which was made according to Habermehl as stated in L102 – 104.
L75: 11 months? See L108.
Answer: Thank you for this, corrected in L 75.
L75-78: Probably these days are selected according the slaughterhouses’ management and number of sheep. I suggest to well define and justify the selection of the sample. Can also the authors insert the official total number of annual reports for sheep slaughters in Switzerland?
Answer: The sampling days were in fact selected according to the slaughterhouses’ management, as sheep and lambs were not slaughtered daily. 260’802 sheep were slaughtered in Switzerland in 2019 (Proviande) (L74 and 75)
L107-108: For M&M
Answer: The amount of included lambs and sheep are mentioned in the Results. We believe this makes sense, as there was no defined amount of sheep and lambs defined in the M&M, but an amount of sampling days.
L117-121: Please don’t repeat results. The figure 1 can be displayed as columns only with the percentage of pregnant ewes. Also apply a non-parametric test to evaluate where the differences between seasons are significant
Answer: The results in the picture are showed to illustrate the difference between the seasons. This is why we believe that the used illustration is helpful for the reader, including the non-pregnant ewes. We have added that pregnancy rates were significantly different between all seasons (p<0.05, based on Chi-square analysis). (line 119)
L123: Why you report the males? The aim is to evaluate the proportion of pregnant females.
Answer: Of course, the reviewer is right. However, this reviewer requested that the total number for sheep slaughters in Switzerland are given. We think that it is important to mention also the number of males, as it gives an information for which percentage of the slaughtered animals the topic is relevant.
I suggest to differentiate pregnant adult from pregnant lamb’s female and test if the differences were significant between them.
Answer: The p-value (based on the Chi-square analysis) is now added. Not significant
Also, why you report each slaughterhouse individually? This is not relevant for this study (the main relevance is if your total sample is representative or not of the total population slaughter during one year in your country).
Answer: we reported every slaughterhouse individually, as in the discussion we argue if the size of the slaughterhouse may correlate with the amount of pregnant slaughtered sheep and lambs. For better understanding though we made a new table without listing every slaughterhouse individually, as the differences between the slaughterhouses were not the main focus of the study. (L132)
L136-143: If you wish to report the single vs multiple pregnancy according to age, please test the data according lamb vs adult and single vs multiple pregnancies (e.g., 4 outcomes groups in a table.
Answer: As the main focus of this study was not set on the single or multiple pregnancies, we don’t consider it necessary to add the statistical relevance of the found data. We just mention the number of found pregnancies to state how many embryos and fetuses were found during the study without any further interpretation.
L144-148: This is mainly right for the 2nd and 3rd third of pregnancy. 150 day /3 = 50: The first right mainly included the embryo phase of pregnancy (not the fetal phase). Please report in M&M this difference in your classification and how to specific detect it (presence of embryo vesicle and embryo).
Answer: As stated in the M&M the estimation of the age of the found embryos and fetuses was carried out basing on the classification scheme of Habermehl, which can be found in the references. Though we did not set a focus on whether the found embryos or fetuses were in the embryo or fetal phase of pregnancy, but on the three pregnancy thirds.
The presence of umbilical cord pulsation and reflexes are dependent of the speed of evaluation. Also, I can understand that the authors insert this evaluation to confirm the ethical of the evaluation; but it was not observed in all the ewes. I think that an average interval between the slaughter and the evaluation should be reported as well as the absence of signs of a non-immediate death of the fetuses (e.g., mummifications/others).
Answer: As mentioned in the M&M the evaluation of the umbilical pulsation or the heartbeat was carried out after the evisceration. The presence of the umbilical cord pulsation was tested in all embryos. The average interval between the slaughter and the evaluation might depend on the slaughterhouse and on the breaks made, therefore we don’t think an average time should be reported. Absence of signs of a non-immediate death of the fetuses were not observed.
L150. Not in trimesters (is not a cow) but estimated in third of gestation.
Answer: Thanks for this comment, we corrected it.
L149-155: Please use an inferential statistical analysis.
Answer: we believe an interferential statistical analysis is not needed, as our goal was to provide descriptive results of the found animals.
L162-167: What is the goal of this information? Do you wish to test differences between regions?
Answer: the aim of this information was to show how many pregnancies were found in which slaughterhouse, to then discuss if the size of the slaughterhouse might have an influence on the prevalence. It was not our goal to test differences between regions.
L169-173: This is a repetition of data.
Answer: We don’t believe that this is a repetition of data, as we for the first time show the prevalence of pregnancies based on the livestock owners and not based on the slaughterhouses. Moreover it is only about the animals slaughtered in autumn. The only repetition of data is that in Autumn 193 of the female animals were pregnant. In our opinion this might be necessary to repeat, as we following list the prevalence of livestock owners responsible for that amount of pregnancies.
L183: 9.6%?
Answer: Thanks for this comment, we corrected it.
L185: You need to test the values. If you wish to use the terms “twice as likely”, please perform an Odds Ratio analysis/presentation in results.
Answer: This sentence was deleted.
L191-198: Attention, for goats birthing at the breeding season, the manifestation of puberty (natural oestrus cycles) only occurs in next breeding season; for several of those occurring during middle and later anestrous season (about May-June to August), this manifestation probably occurs in the second breeding season (about 14-18 month later) when they are mated. Please turn clearer this part of the discussion, and add some additional references, once they can occur a great variability.
Answer: We stated from which age on lambs can manifest puberty, based on the given source. This was to emphasize, that even if they are called lambs in the slaughterhouse, they might be pregnant at slaughter. It was not further part of the discussion to state when they mostly get pregnant, as the main aim was to show that there is the possibility for pregnancies from an early age on.
L195: The most part of the males reach puberty at 6-8 month, no? Please see the comment for females.
Answer: The used reference stated, that puberty in rams can already occur at an age of 3 to 6 months.
L200-203: Please see my comment regarding the OR.
Answer: Thanks for this comment. This part was rewritten and the p-value is now given.
L212-217: About 12 farmers presented more than 10%. One doubt: during non-breeding season, what is the percentage of sheep that can ovulate naturally? Or these ewes were subjected to estrous synchronization? Probably a discussion about this aspect can better explain some of your results.
Answer: Thanks for this comment. No information regarding the ovulation of the slaughtered sheep was provided, nor was it stated if the ewes were subjected to estrous synchronization.
L218-226: “Having a lower number of animals might result in more accurate awareness of their pregnancy”. The differences between slaughterhouses B e C are not evident.
Answer: Thanks for this comment. Our statement was just a possible, general reason for less pregnancies in smaller slaughterhouses and not an evident claim.
L228-244: I don’t understand why you discuss this subject. You need to well define the estimation of the 1st, 2nd and 3rd pregnancy period in M&M. Some of this present information can be considered at definition time.
Answer: As stated the estimation of the gestational age was performed by Habermehl. More information can be found in the source. We discuss the object as also other sources might be possible for age estimation which could possibly have another outcome.
L246: “second pregnancy trimester” is not a trimester. Please correct in whole the text.
Answer: Thanks for this comment, we corrected it in the whole text.
L245-253: I think that is important to discuss the pregnancy diagnosis in flock (slaughterhouses?) by echography, once is a quick and easy tool. In the 3rd pregnancy period, the pregnancy also can be diagnosed by abdominal succussion.
Answer: The main aim of this study was not the pregnancy diagnosis, but the data collection of pregnant ewes and ewe lambs, found pregnant after opening the uterus. Therefore, the pregnancy diagnosis was not further discussed.
L257: “…that the older a sheep is, …”. Sorry but no entirely true: nulliparous ewes carry mainly single fetus; primiparous and multiparous females have more proportion of twins/multiple fetuses up to an old age/parity limit. The real (relevant) differences are between nulliparous and other parities.
Answer: Thanks for this comment, we corrected it.
L266-268: The interdigital reflex is a defense mechanism. I think that is relevant to report the interval between the slaughter of the mother and the evaluation of the fetus. Once the fetus is deprived of O2 and accumulate CO2, a progressive fetal depression occurs.
Answer: Thanks for this comment. The fetal depression was mentioned in the discussion and also the interdigital reflex being a defense mechanism.
L300: A bovine fetus?
Answer: No, an ovine fetus. This was added to the text.
L325-326: Right. And at the current technology to make an easy and economic pregnancy diagnosis. There is no justification to slaughter pregnant females. It is nor ethical neither environmental (direct and indirect energy cost of a pregnancy) appropriate.
Answer: Thank you for this comment.
L331: The higher proportion.
Answer: Thanks for this comment, we corrected it.
L336-338: These preventive measures fit better if discussed in a subsection of the discussion section.
Answer: In our opinion the preventive measures are well placed in the conclusions, as it was not the aim to mention them further in the discussion.
Round 2
Reviewer 2 Report
Dear authors,
Thanks for providing this revised version. All comments and suggestions were adressed or appropriately /with justification) refuted.
I have just to additional issues:
1- L103-104. The reference [8] (Habermehl, 1975) was published in germany language. I suggest to quickly describe his classification in the manuscript. This can help further researches.
2- I also suggest to add a subsection in M&M regarding the statistical analysis (descriptive analysis plus chi-square testing diferences between groups).
Author Response
1- L103-104. The reference [8] (Habermehl, 1975) was published in germany language. I suggest to quickly describe his classification in the manuscript. This can help further researches.
Answer: is now added “Briefly, with a crown-rump length between 5 centimeters and 27 centimeters and as soon as the gender can be identified the fetuses are classified in the second pregnancy third, whereas if the crown rump length is 27 cm or more and the Sinus inguinalis is developed, the fetuses are classified in the third pregnancy third.”
2- I also suggest to add a subsection in M&M regarding the statistical analysis (descriptive analysis plus chi-square testing differences between groups).
Answer: a new section 2.3. Statistics is now added: Descriptive analysis was used to summarize the collected data. Significance of differences between groups were identified by chi-square test using GraphPad Prism (Version 9.3.0 (345), GraphPad Software, San Diego, CA, USA), considering p values < 0.05 to be statistically significant.